# Simple Nearest Neighbor Policy Method for Continuous Control Tasks

## Abstract

We design a new policy, called a nearest neighbor policy, that does not require any optimization for simple, low-dimensional continuous control tasks. As this policy does not require any optimization, it allows us to investigate the underlying difficulty of a task without being distracted by optimization difficulty of a learning algorithm. We propose two variants, one that retrieves an entire trajectory based on a pair of initial and goal states, and the other retrieving a partial trajectory based on a pair of current and goal states. We test the proposed policies on five widely-used benchmark continuous control tasks with a sparse reward: Reacher, Half Cheetah, Double Pendulum, Cart Pole and Mountain Car. We observe that the majority (the first four) of these tasks, which have been considered difficult, are easily solved by the proposed policies with high success rates, indicating that reported difficulties of them may have likely been due to the optimization difficulty. Our work suggests that it is necessary to evaluate any sophisticated policy learning algorithm on more challenging problems in order to truly assess the advances from them.

## 1 Introduction

Model-free reinforcement learning for continuous control tasks have received renewed interest in recent years (see, e.g., Arulkumaran et al., 2017). Most of recently proposed algorithms rely on learning a parametrized policy that takes as input an observation, or a sequence of observations, and outputs a real-valued action. This process of learning a parametrized policy can be understood as lossy compression. During training, the (noisy) policy is executed multiple times to collect a set of trajectories. These trajectories are then compressed into a fixed number of parameters of the policy, while ensuring that the policy would eventually generate a trajectory that solves the task.

Under this view of and approach to policy learning for continuous control, we notice that there are two distinct factors contributing to the success of a policy learning algorithm. First, as usual with any reinforcement learning scenario, the difficulty of an environment and task directly influences the outcome of a learning algorithm. Second, the effectiveness of compressing a set of collected trajectories into a set of parameters defining a policy either positively or negatively affects the overall performance of a learned policy. In other words, the success or failure of a learned policy cannot be attributed to any one of these two factors, but is often due to the mix of these two factors–task difficulty and optimization difficulty. As a consequence, the success or failure of a policy learning algorithm cannot be taken as an indicator of how challenging a task is. The failure of a learned policy may be entirely due to optimization alone.

In this paper, we attempt to study existing continuous control tasks widely used for benchmarking policy learning algorithms by separating out those two factors. We do so by devising a new type of policy based on nearest neighbor retrieval. This new family of policies, called *nearest neighbor policy methods*, skips the entire step of optimization or lossy compression by maintaining a buffer of successful trajectories from exploration. When executed, the nearest neighbor policy retrieves a trajectory from the buffer based on the current or initial state and goal state. The action from the retrieved trajectory is taken as it is. Because this family of policy does not suffer from challenges in optimization, its success or failure more closely reflects the true difficulty of an environment and exploration in the task.

With the proposed nearest neighbor policy, we evaluate five widely-studied, low-dimensional continuous control tasks; Reacher, Half-Cheetah, Cart Pole, Double Pendulum and Mountain Car. We choose a sparse reward function (+1 if successful, and otherwise 0) to ensure that these tasks are challenging. Our experiments reveal that the proposed nearest neighbor policies easily solve Reacher, Half Cheetah, Cart Pole and Double Pendulum with high success rates, while they fail on Mountain Car. This suggests that a majority of these widely used tasks are not adequate for benchmarking the advances in model-free reinforcement learning for continuous control tasks, and that more challenging tasks and environments must be considered in order to truly assess the effectiveness of new algorithms on their ability to solve difficult tasks.

Despite the success of these simple nearest neighbor policies, our further analysis reveals some aspects that are worth discussion. First, we observe that the proposed policies find strategies that are not smooth, thus less natural and adequate for real-world applications. Second, we observe that both variants of the nearest neighbor policy deteriorate with noisy actuation during the test time. This suggests that more investigation into how to learn or design a robust distance function is necessary in the future.

## 2 QUICK RECAP: POLICY GRADIENT

Let us denote by $\pi$ a policy that maps from an observation $x \in \mathbb{R}^d$ to an action $a \in \mathbb{R}^k$. The goal of policy gradient is to find a policy $\pi$ that maximizes the expected return $R$, or the sum of per-step rewards $r_t$, over all possible trajectories according to the policy. That is,

$$\hat{\pi} = \arg \max_{\pi} \mathbb{E}_{\mathcal{T} \sim \pi} \left[ R(\mathcal{T}) \right], \tag{1}$$

where $\mathcal{T} = ((x_1, a_1, r_1), \ldots, (x_T, a_T, r_T))$, and $a_t \sim \pi(x_{\leq t})$. When $\pi$ is a deterministic function, sampling an action $a_t$ is often done by adding unstructured noise $\epsilon$ to the policy's output, i.e., $a_t = \pi(x_{\leq t}) + \epsilon_t$, where $\epsilon_t \sim \mathcal{OU}(0, \sigma^2)$, where $\mathcal{OU}$ is Ornstein-Uhlenbeck process (Lillicrap et al., 2015).

**REINFORCE** It is almost always intractable to exactly evaluate the objective function in Eq. (1). Most importantly, the return, or the sum of per-step rewards, is not differentiable with respect to the policy $\pi$, preventing us from using a gradient-based optimization algorithm. This issue is often circumvented by a likelihood ratio method, such as REINFORCE (Williams, 1992). Together with Monte Carlo approximation, we end up

$$\nabla_{\pi} \mathbb{E}_{\mathcal{T} \sim \pi} \left[ R(\mathcal{T}) \right] \approx \sum_{t=1}^{T} \left( \sum_{t'=t}^{T} r_{t'} \right) \nabla \log \pi(a_t | x_{\leq t}).$$

In order to reduce the variance of this estimator, it is a usual practice to subtract a baseline $b_t$ from the cumulative future reward, resulting in

$$\nabla_{\pi} \mathbb{E}_{\mathcal{T} \sim \pi} \left[ R(\mathcal{T}) \right] \approx \sum_{t=1}^{T} \left( \sum_{t'=t}^{T} r_{t'} - b_t \right) \nabla \log \pi(a_t | x_{\leq t}). \tag{2}$$

The baseline is often set to an expected cumulative future reward, or a value given the state $x_{\leq t}$. This estimated gradient can be used with simple stochastic gradient descent or with a more sophisticated optimization algorithm, such as trust-region policy optimization (Schulman et al., 2015; Wu et al., 2017), either via Fisher-matrix vector products Martens (2010) or Kronecker-factored trust region (Martens & Grosse; Grosse & Martens, 2016; Ba et al., 2017).

Updating the policy $\pi$ according to the learning rule above is equivalent to the following strategy. First, we obtain a sample trajectory from the policy by sampling from the action distribution at each time step. For each step $t$ in the trajectory, if the cumulative future reward $\sum_{t'=t}^{T} r_{t'}$ is larger than the estimated expected one $b_t$, we increase the probability assigned to the selected action $a_t$. Otherwise, we decrease it. In other words, we encourage the policy to select the action if the action was found to lead to a higher future reward, and vice versa.

**Deep Deterministic Policy Gradient (DDPG)**    Another widely used approach is DDPG (Lillicrap et al., 2015). Instead of directly maximizing the objective function in Eq. (1) with respect to a policy $\pi$, this algorithm first builds a parametrized critic $c_\phi(s, a)$ that approximates the expected cumulative future reward $Q(s, a)$. This critic acts as a differentiable proxy to the true objective function. This differentiability allows us to estimate the gradient of the objective function with respect to the policy by

$$\nabla_\pi c_\phi(s, \pi(s)) = \nabla_a c_\phi(s, a) \nabla_\pi \pi(s),$$

given a collected trajectory.

**Goal**    Ultimately, the goal of learning such a policy, regardless of which algorithm was selected, is to find a policy $\hat{\pi}$ that almost always generates a trajectory $\mathcal{T}^*$ that leads to the maximal return. This search for the policy is based on a set of generated trajectories $D = \{\mathcal{T}_1, \ldots, \mathcal{T}_M\}$, to which we refer as a function $\mathcal{M}$. That is, $\hat{\pi} = \mathcal{M}(D)$.

In a usual, parametric case, this process of learning maps from the set of trajectories $D$ to a fixed set of parameters $\theta$ specifying a policy $\pi_\theta$ which in turn generates the optimal trajectory $\mathcal{T}^*$. This is however not the only option, and we may resort to a non-parametric approach. In the non-parametric approach, the entire process $\mathcal{M}$ maps from $D$ to the optimal trajectory $\mathcal{T}^*$ directly. There may be some optional set of parameters inferred from $D$ in an intermediate stage, but the inference of the optimal trajectory is largely driven by the original set $D$. This latter approach is what we focus on this paper.

**Why (not) a parametrized policy?**    The prevalence of parametrized policies is due to two major reasons. First, it is computationally more efficient in test time. The size of the policy, roughly equivalent to the number of parameters, is constant with respect to the number of trajectories collected during training. Furthermore, this property of compression allows the policy to generalize better to unseen states.

Despite these advantages, there also are downsides to the parametric approach to policy learning. Most of these downsides arise from the necessity of compressing trajectories collected during training into the fixed set of parameters. Compression is done by optimization, and this optimization process could be difficult, regardless of the difficulty of the target task (Henderson et al., 2017). This implies that the success or failure of learning a parametrized policy depends on two factors. They are the difficulty of the target task and the difficulty in optimization. In other words, the performance of a final policy does not directly indicate the difficulty of a target task.

This observation raises a question of which one of these two aspects each claim on advances in policy learning is on. Without being able to tell apart the contributions from these two aspects to the final performance, can we tell whether a novel policy learning algorithm allows us to tackle a more difficult problem? In this paper, we attempt at at least indirectly answer this question by considering a policy learning algorithm that does not rely on compression, or in other words, is not a parametrized policy.

## 3    NEAREST NEIGHBOR POLICY METHODS

There have been a number of non-parametric approaches to reinforcement learning in recent years (Pritzel et al., 2017; Blundell et al., 2016; Rajeswaran et al., 2017; Lee & Anderson, 2016). Most of these have focused on learning a $Q$ function of which size grows with respect to the number of generated trajectories (and corresponding state-action-reward pairs.) This allows the $Q$ function to adapt its complexity flexibly according to the amount of collected trajectories and the associated complexity, unlike the parametric counterpart in which the maximum capacity is capped by the fixed number of parameters describing the $Q$ function. Most of these approaches however have a substantial number of free parameters that must be estimated during training.

In this paper, we propose a similar approach directly to learning a policy. More specifically, we design a policy learning algorithm that does *not* require any parameter estimation.[1] The proposed

---

[1] We however illustrate later how the proposed approach could be extended to have free parameters estimated from learning.

approach is an adaptation of a nearest neighbor classifier to reinforcement learning. Below, we describe two variants of the nearest neighbor (NN) policy.

**Notations**    Let us first define notations.

- $D$: a set of all collected trajectories
- $B \subseteq D$: a buffer storing a subset of all trajectories
- $s_0$: an initial state (initial state consists of initial positions of joints, initial velocities + distance to goal object (if there is a goal in the environment like in Reacher)
- $d(\cdot, \cdot)$: a distance function
- $\tau$: a return threshold

**NN-1**    The first variant works at the level of an entire trajectory. At the beginning of each episode, this policy (**NN-1**) looks up the nearest neighboring trajectory $\hat{\mathcal{T}}$ from the buffer $B$. The lookup is based on the initial state $s_0^*$ in the current episode. That is,

$$\hat{\mathcal{T}} = \arg \min_{\mathcal{T} \in B} d(s_0^*, s_0)$$

where $s_0$ is the initial state associated with a trajectory $\mathcal{T}$ stored in the buffer $B$. In other words, we use the initial state of a stored trajectory as a key to find the nearest neighboring trajectory. Once the trajectory has been retrieved, the NN-1 policy executes it by adding noise $\epsilon$ to each retrieved action $\hat{a}_t$. When $B$ is empty, we simply sample actions uniformly at each timestep. When evaluating the NN-1 policy, we simply set the variance of noise to 0.

Once the episode terminates resulting in a new trajectory $\mathcal{T}^*$, we store it in the buffer, if the return $\sum_{t=1}^{|\mathcal{T}^*|} r_t^*$ is larger than a predefined threshold $\tau$:

$$B \leftarrow B \cup \left\{ (s_0^*, \mathcal{T}^*) \right\}, \text{ if } \sum_{t=1}^{|\mathcal{T}^*|} r_t^* \geq \tau.$$

**NN-2**    Instead of retrieving the actions for the entire episode, the second variant, called **NN-2**, stores a tuple of state $s_t$, action $a_t$ and cumulative reward $\bar{r}_t = \sum_{t'=t}^{T} r_{t'}$. Thus, retrieval from the buffer happens not only at the beginning of the episode, but at every step along the trajectory. At each step,

$$\hat{a}_t = \arg \min_{a_{t'}} d(s_t^*, s_t).$$

During training, unstructured noise is added to the retrieved action $\hat{a}_t$, similarly to the NN-1. When evaluating this policy, the variance of the noise is set to 0.

Once the episode completes, we store all the tuples $(g^*, s_t^*, a_t^*, \bar{r}_t^*)$ from the episode into the buffer $B$, if the full return is above the threshold $\tau$:

$$B \leftarrow B \cup \left\{ (g^*, s_t^*, a_t^*, \bar{r}_t^*) \in) \right\}, \text{ if } \sum_{t=1}^{|\mathcal{T}^*|} r_t^* \geq \tau.$$

Despite the similarity between the NN-1 and NN-2, there is an important difference. In the evaluation time, the NN-1 policy can only generate a trajectory stored in the buffer, while the NN-2 policy may return a novel trajectory created by weaving together pieces of the trajectories in the buffer. This allows the NN-2 to better cope with unexpected stochastic behaviors during execution. This advantage however comes with a higher computational load due to repeated nearest neighbor retrieval.

**What does the NN policy do?**    Both variants of the proposed NN policy precisely learn a policy that would be learned by any policy gradient algorithm, in the case of a continuous action. As described in the earlier section, the goal of policy gradient is for a learned policy to generate an optimal trajectory based on a set of collected trajectories. A stark difference from the conventional, parametric approach is that there is no compression of collected trajectories into a fixed set of parameters.

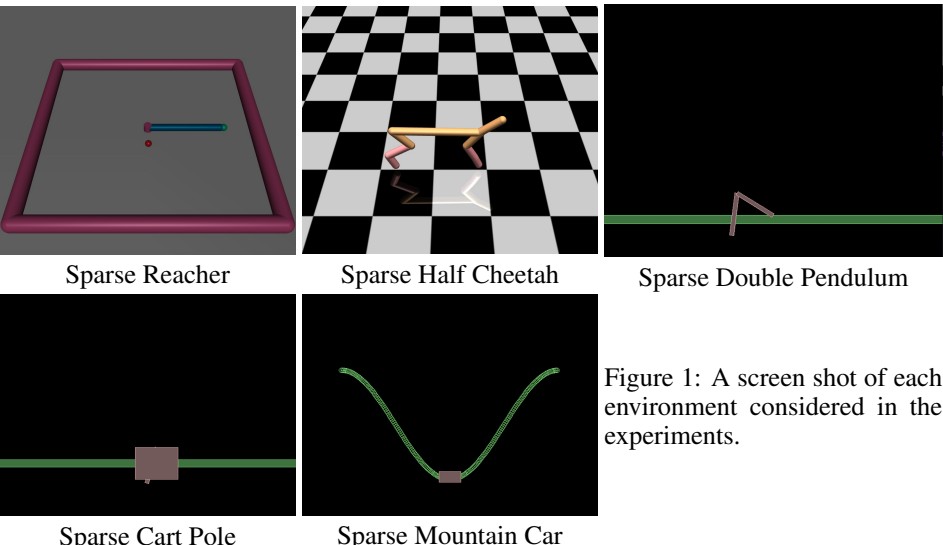

| Sparse Reacher | Sparse Half Cheetah | Sparse Double Pendulum |

| Sparse Cart Pole | Sparse Mountain Car |

Figure 1: A screen shot of each environment considered in the experiments.

This lack of compression frees the proposed nearest neighbor policy algorithms from actual *learning*. Instead, the nearest neighbor policy requires the retrieval procedure whose complexity grows with respect to the size of the buffer. Also, the choice of the distance function $d$ must be made carefully.

**Trainable NN Policy**  There are two aspects of the proposed policies that could be estimated by learning rather than predefined in advance. The first one is the distance function $d$. Following the earlier approaches such as the neural episodic control (Pritzel et al., 2017), it is indeed possible to learn a parametrized distance function $d_\theta$ rather than pre-specifying it in advance. The second one is the threshold $\tau$. The threshold could be adapted on the fly to encourage gradual exploration of different strategies. Since our aim in this paper is to understand the underlying complexity of existing, simple robotics tasks, we leave the investigation of adapting these aspects to the future. However, in the experiments, we demonstrate what kind of effect the choice of the threshold has on the final policy.

**Connection to tabular Q learning**  Tabular Q learning is another, more traditional approach to reinforcement learning without compression. Instead of carrying the entire trajectory set $D$ or compressing it into a fixed set of parameters $\theta$, tabular Q learning transforms it into another form resembling a table. The rows of this table correspond to all possible states $s_{\leq t}$, and the columns all possible actions $a_t$. Each cell contains the quality of the associated pair $(s_{\leq t}, a_t)$. At each step, the row corresponding to the current state is consulted, and the action with the highest quality value $Q$ is selected. This approach is attractive, because it does not involve forced compression of $D$. It however has a number of weaknesses, two of which are the lack of scalability and the lack of a principled way to handle continuous state and action.

## 4 EXPERIMENT SETTINGS

**Environments**  Let us restate our goal in this paper. We aim at determining the adequacy of popular simulated environments for robotics control tasks by investigating the performance of the proposed nearest neighbor policies on these environments. In order to achieve this goal, we test the following five tasks in an environment simulated using MuJoCo (Todorov et al., 2012) and Box2D physics engines[2] that can be found in (Houthooft et al., 2016) and shown in Fig. 1:

- **Sparse Reacher**: A 2-D arm has to reach a target. The reward of $+1$ is given if $\|\text{pos}_{\text{fingertip}} - \text{pos}_{\text{target}}\| \leq 0.04$, otherwise reward of $0$ is given. $s \in \mathbb{R}^{11}$ and $a \in \mathbb{R}^2$.
- **Sparse Half-Cheetah**: The goal is to move a 6-DOF cheetah forward. The reward of $+1$ is given when $\text{pos}_{\text{torso}} \geq 5$. $s \in \mathbb{R}^{17}$ and $a \in \mathbb{R}^6$.

---

[2] https://github.com/erincatto/Box2D

- **Sparse Cartpole Swingup**: The reward of $+1$ is given if $\cos(\text{angle}_{\text{pole}}) > 0.8$. $s \in \mathbb{R}^4$ and $a \in \mathbb{R}^1$.

- **Sparse Double Pendulum**: The reward of $+1$ is given if the $\|\text{pos}_{\text{fingertip}} - \text{pos}_{\text{target}}\| \leq 1$, where the target is a straight arm. $s \in \mathbb{R}^6$ and $a \in \mathbb{R}^1$.

- **Sparse Mountain Car**: The reward of $+1$ is given if mountain car reaches the goal position of 0.6. $s \in \mathbb{R}^2$ and $a \in \mathbb{R}^1$.

In all the cases, we consider only a sparse reward (1 if successful and 0 otherwise) to ensure that we are considering the most sophisticated setting given an environment. Despite the simplicity of the environment and associated task, we notice that many existing policy gradient methods, such as DDPG, have been found to fail unless equipped with an elaborate exploration strategy (see, e.g., Plappert et al., 2017; Houthooft et al., 2016).

**Evaluation Metrics**  Because we test only on sparse environments, the main evaluation metric is the **success rate**, that is, the number of successful runs over all the evaluation runs. In order to assess the efficiency of the proposed algorithms, we also measure the **number of iterations** necessary until the success rate reaches a predefined threshold. Since there is no process of learning (i.e., compression), these two metrics directly measure the difficulty, or its upperbound,[3] of an environment and task.

In addition to these metrics, which are standard, we also evaluate the perceived quality of a final policy. This is importance because there are often many different strategies that could solve a task in a simulated environment. Many of these strategies however will not be applicable or usable in a real environment due to their extreme behaviors. We thus consider the **average norm of the action** which should ideally be small.

**Algorithm Settings**  We test both NN-1 and NN-2 on all the tasks. Since these tasks are of a binary, sparse reward, we simply set the threshold $\tau$ to 1, unless otherwise specified. We use Euclidean distance for nearest neighbor retrieval, i.e., $d(x,y) = \|x - y\|_2$. During training, we sample noise $\epsilon$ from Ornstein-Uhlenbeck process with $\sigma = 0.2$, as we found it to be superior over normal distribution in the preliminary experiments. For the NN-2 approach, we use the concatenation of the past three observations and the current observation for retrieval. In order to avoid the growing computational overhead from retrieval, we set the maximum size of the buffer, if necessary.

## 5   RESULT AND ANALYSIS

### 5.1   PERFORMANCES OF THE NN-1 AND NN-2 POLICIES

As shown in Fig. 2, in all but one task (Sparse Mountain Car), the proposed NN policies, or at least one of them, rapidly succeeds in solving them with only a small number of trajectories collected. On Sparse Reacher and Sparse Double Pendulum, both of the proposed policies solve the problems with more than 90% success rate. On Sparse Cart Pole, the NN-2 successfully solves the task with near 100% success rate, when each action was issued four times at time step to the environment. Both the NN-1 and NN-2 policies achieve more than 70% success rate on Sparse Half Cheetah which is considered very difficult to solve without a sophisticated exploration strategy using a parametrized policy (see, e.g., Houthooft et al., 2016; Plappert et al., 2017). This happened even with the limitation on the size of the buffer when the NN-2 policy was used. On the other hand, neither the NN-1 nor NN-2 policy was able to solve Sparse Mountain Car, which indicates the inherent difficulty of the task itself and that a better algorithm, such as a more sophisticated exploration strategy (see, e.g., Houthooft et al., 2016).

### 5.2   PERCEIVED QUALITY OF THE NEAREST NEIGHBOR POLICIES

---

[3] This is an upperbound on the difficulty, as the generalization capability of a parametrized policy may allow it to solve the task more easily.

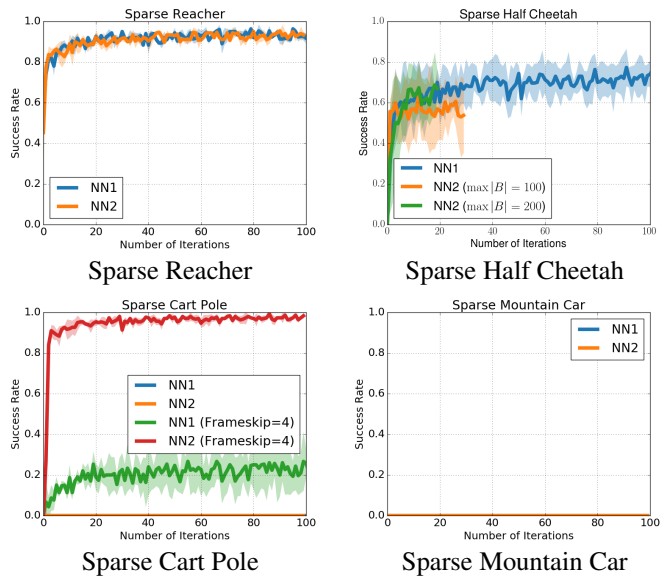

Sparse Reacher

Sparse Half Cheetah

Sparse Double Pendulum

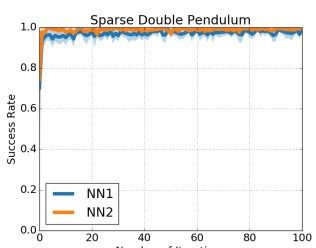

Figure 2: The success rate of each environment over trajectory collection. At each iteration we evaluate the performance of NN policies on 100 episodes. Each iteration consist of 100 training episodes.

Sparse Cart Pole

Sparse Mountain Car

On inspecting the final NN-1 policy on Reacher, we notice that the strategy found by the policy is not smooth. In the case of Half Cheetah, we observe the cheetah walking forward only until it reaches the goal and starting to walk backward. These observations shed light on a weakness of the proposed policies. That is, it is not easy to directly incorporate penalties on their behavior. This is in contrast to optimization-based approaches, where it is natural to add in regularization terms to encourage desired behavior, such as a smooth trajectory.

In order to quantitatively compare the proposed policies against parametrized policies, we train a parametrized policy for each task, however, with a dense reward using ACKTR (Wu et al., 2017).[4] When training the parametrized policies, we explicitly penalize the norm of each action for Reacher and Half Cheetah, as done in OpenAI Gym (Brockman et al., 2016). This encourages them to learn a smooth trajectory. Such per-step reward shaping cannot be applied to the proposed nearest neighbor policies.

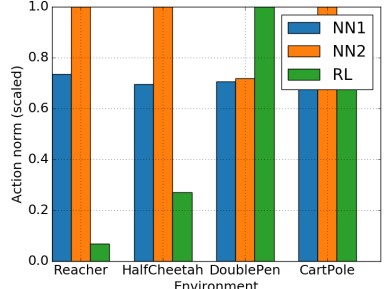

Figure 3: The action norms of the proposed NN-1 and NN-2 policies as well as the parametrized policy. For each task, the action norms were scaled so that the largest one is 1. RL refers to the parametrized policies.

We then plot the average action norms of these policies: NN-1, NN-2 and the parametrized policy in Fig. 3. The average action norm can be made artificially small in the case of parametrized policy, as evident from Reacher and Half Cheetah. This is however not possible with the proposed NN-1 and NN-2 policies, resulting in less stable, albeit successful, trajectories. When we do not explicitly encourage such smooth behavior, the action norms do not differ much between the nearest neighbor and parametrized policies.

This reveals two things. First, the current formulation of the nearest neighbor policy does not allow us to explicitly encourage a specific property of a trajectory from the policy. This should be investigated further in the future. Second and perhaps more important, the evaluation of a policy, either parametrized or nearest neighbor one, needs to take into account not only the success rate but also perceived quality.

### 5.3 EFFECT OF THE THRESHOLD WITH AN UNBOUNDED CUMULATIVE REWARD

Although it is sensible to set the threshold $\tau$ to the maximum return in the case of sparse reward environments, it may be desirable to set it to an arbitrary value, when the policy can accumulate

---

[4] Note that we were not able to train a parametrized policy with a sparse reward on any of the environments.

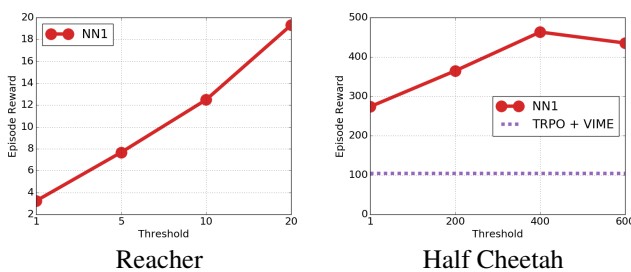

Figure 4: The average returns by the NN-1 policy with different values of threshold $\tau$ on Reacher and Half Cheetah with dense rewards. In the case of Half Cheetah, as a reference, we use the dashed line to indicate the return achieved by a parametrized policy reported recently by Houthooft et al. (2016).

rewards indefinitely as it executes in the environment. The choice of the threshold has a direct influence in the behavior of the proposed nearest neighbor policies, as it sets a higher bar on which of the collected trajectories are saved in a buffer. Here we thus test the effect of the threshold on two tasks–Reacher and Half Cheetah– under a dense reward situation. In other words, the policy would receive a non-zero reward each time it successfully continues its execution without a failure.

In Fig. 4, we plot the relationship between the threshold and the final cumulative reward per task achieved by the NN-1 policy. We observe that it is generally possible for us to control the quality of the NN-1 policy by adjusting the threshold. However, as can be noticed from the result on Half Cheetah, the performance does not necessarily increase proportionally to the threshold. We believe this is due to the issue of sparsity. That is, the coverage of the state space decreases as the length of allowed episodes grows, and the nearest neighbor retrieval cannot generalize well to unseen states. Despite this, we notice that the average return of up to 450 achieved by the NN-1 policy is higher than that achieved by a parametrized policy trained with an advanced policy learning algorithm (Houthooft et al., 2016) (marked by a dashed line.)

This observation suggests a future extension of the proposed nearest neighbor policies that automatically adjusts the threshold. This will necessarily require pruning trajectories from the buffer. As such an extension, which likely requires optimization, is out of the scope of this paper, we leave this for the future.

## 5.4 TRANSFERABILITY TO A NOISY ACTUATOR

As the proposed approach is essentially based on memorization of collected trajectories, it is valid to be concerned of its robustness when the environment is noisy. In this section, we analyze the robustness of the nearest neighbor policy by simulating a noisy actuator during the test time. That is, we obtain a nearest neighbor policy with a clean actuator, and test it with a noisy actuator. The noisy actuator is simulated by having sticky action (Machado et al., 2017). We vary the chance of skipping the current action among 5%, 25% and 50%. We test both NN-1 and NN-2 policies on Reacher under the sparse reward configuration.

The success rates under noisy actuation during training are presented in Table 1. As expected, the success rate deteriorates as the amount of noise increases. In the case of Reacher, both of the policies however maintain the success rate nearly 50% even when the chance of ignored action is half. We conjecture that this deterioration is largely due to the sensitivity of our choice of the distance function to noise, implying that there is a room for further improvement in the nearest neighbor policy.

## 6 CONCLUSION

In order to assess the underlying difficulties of widely-used tasks in model-free reinforcement learning for continuous control, we designed a novel policy, called the nearest neighbor policy, that does not require any optimization, which allows us to avoid any difficulty from optimization. We evaluated two variants, NN-1 and NN-2, of the proposed policy on five tasks–Reacher, Half Cheetah, Double Pendulum, Cart Pole and

| Task | Policy | Noise Level | | | |
|---|---|---|---|---|---|
| | | 0% | 5% | 25% | 50% |
| Reacher | NN-1 | .96 | .76 | .61 | .51 |
| | NN-2 | .94 | .73 | .55 | .49 |

Table 1: The success rates of the NN-1 and NN-2 policies in the environment with noisy actuation.

Mountain Car– which are known to be challenging especially with a sparse reward. Despite the simplicity of the proposed policy, we found that one or both of the variants solved all the environments except for Mountain Car. This suggests that the perceived difficulty of these benchmark problems has largely been dominated by optimization difficulty rather than the actual, underlying difficulty of the environment and task. From this observation, we conclude that more challenging tasks, or more diverse settings of existing benchmark tasks, are necessary to properly assess advances made by sophisticated, optimization-based policy learning algorithms.

**Future directions for the nearest neighbor policy**  The proposed nearest neighbor approach was deliberately made as simple as possible in order to remove any need of optimization. This however does not imply that it cannot be extended further. As has been done for learning a $Q$ function in recent years (Pritzel et al., 2017; Blundell et al., 2016; Rajeswaran et al., 2017; Lee & Anderson, 2016), we can put a parametrized function that learns to fuse between multiple retrieved trajectories. A similar approach has recently been tried in the context of neural machine translation by Gu et al. (2017). Also, as mentioned earlier in Sec. 3, the threshold and the maximum size of the buffer may be adapted on-the-fly, and this adaptation policy can be learned instead of manually designed. This last direction will be a necessity to overcome the curse of dimensionality in order to apply the nearest neighbor policy to tasks with a high-dimensional observation and action.

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
