# OpenReview forum: "Simple Nearest Neighbor Policy Method for Continuous Control Tasks"
_ICLR.cc/2018/Conference — Reject_

### Official Review · AnonReviewer3 · 2017-11-17
**The algorithm presented despite its simplicity cannot compete with classic RL algorithms.  The claim about assessing problem difficulty with this approach is not well justified.**

**Rating:** 4
**Confidence:** 5

**Review:**

SUMMARY
The paper deal with the problem of RL.  It proposes a non-parametric approach that maps trajectories to the optimal policy.  It avoids learning parameterized policies.  The fundamental idea is to store passed trajectories.  When a policy is to be executed, it does nearest neighbor search to find then closest trajectory and executes it.

COMMENTS

What happens if the agent finds it self  in a state that while is close to a state in the similar trajectory the action required to could be completely different.

Not certain about the claim that standard RL policy learning algorithms make it difficult to assess the difficulty of a problem.

How do you execute a trajectory? Actions in RL are by definition stochastic, and this would make it unlikely that a same trajectory can be reproduced exactly.

---

> ### Author Response · Authors · 2018-01-05
> **author response**
>
> "What happens if the agent finds itself  in a state that while is close to a state in the similar trajectory the action required to could be completely different."
>
> This would be indeed an issue with any nearest neighbor based policy, and the two variants in our submission would also suffer from this issue. However, we'd like to point out that the success of these variants on the four out of five tasks we've tried in this submission suggests that such a situation does not naturally/frequently occur in these tasks, leading us to conclude that these tasks are not adequate for evaluating any sophisticated policy parametrizations nor learning algorithms. This is our main conclusion and motivation in this submission.
>
> "Not certain about the claim that standard RL policy learning algorithms make it difficult to assess the difficulty of a problem."
>
> We believe we have explained it at the end of Sec. 2. It would be better if you could share with us which part of our explanation you find less certain of. We would like to improve our submission.
>
> "How do you execute a trajectory?"
>
> As stated in Sec. 3, "[o]nce the trajectory has been retrieved, the NN-1 policy executes it by adding noise ε to each retrieved action".
>
> "Actions in RL are by definition stochastic"
>
> We also do not believe this is necessarily true. In the case of deterministic policy, actions are stochastically corrupted during training for exploration, but the underlying policy could as well be deterministic.

---

### Official Review · AnonReviewer2 · 2017-11-27
**Limited impact**

**Rating:** 4
**Confidence:** 4

**Review:**

This work shows that a simple non-parametric approach of storing state embeddings with the associated Monte Carlo returns is sufficient to solve several benchmark continuous control problems with sparse rewards (reacher, half-cheetah, double pendulum, cart pole) (due to the need to threshold a return the algorithms work less well with dense rewards, but with the introduction of a hyper-parameter is capable of solving several tasks there). The authors argue that the success of these simple approaches on these tasks suggest that more changing problems need to be used to assess new RL algorithms.

This paper is clearly written and it is important to compare simple approaches on benchmark problems. There are a number of interesting and intriguing side-notes and pieces of future work mentioned.

However, the originality and significance of this work is a significant drawback. The use non-parametric approaches to the action-value function go back to at least [1] (and probably much further). So the algorithms themselves are not particularly novel, and are limited to nearly-deterministic domains with either single sparse rewards (success or failure rewards) or introducing extra hyper-parameters per task.

The significance of this work would still be quite strong if, as the author's suggest, these benchmarks were being widely used to assess more sophisticated algorithms and yet these tasks were mastered by such simple algorithms with no learnable parameters. Yet, the results do not support the claim. Even if we ignore that for most tasks only the sparse reward (which favors this algorithm) version was examined, these author's only demonstrate success on 4, relatively simple tasks.

While these simple tasks are useful for diagnostics, it is well-known that these tasks are simple and, as the author's suggest "more challenging tasks  .... are necessary to properly assess advances made by sophisticated, optimization-based policy algorithms." Lillicrap et al. (2015) benchmarked against 27 tasks, Houtfout et al. (2016) compared in the paper also used Walker2D and Swimmer (not used in this paper) as did [2], OpenAI Gym contains many more control environments than the 4 solved here and significant research is pursing complex manipulation and grasping tasks (e.g. [3]). This suggests the author's claim has already been widely heeded and this work will be of limited interest.

[1] Juan, C., Sutton, R. S., & Ram, A. Experiments with Reinforcement Learning in Problems with Continuous State and Action Spaces.

[2] Henderson, P., Islam, R., Bachman, P., Pineau, J., Precup, D., & Meger, D. (2017). Deep reinforcement learning that matters. arXiv preprint arXiv:1709.06560.

[3] Nair, A., McGrew, B., Andrychowicz, M., Zaremba, W., & Abbeel, P. (2017). Overcoming exploration in reinforcement learning with demonstrations. arXiv preprint arXiv:1709.10089.

---

> ### Public Comment · (anonymous) · 2017-12-30
> **Followup comment**
>
> Disclosure: not an author on this paper
>
> While I welcome the comment that only a limited number of tasks have been studied in this paper, I would like to ask if the same standards are adhered more broadly in the community. While the reviewers seem to acknowledge that these tasks are only toy examples, there seems to be a flurry of papers that propose and evaluate *novel* algorithms only on these tasks. There seems to be a big discrepancy here, with these novel approaches trying to kill a mosquito with a revolver.
>
> I would like to raise a more general question to both this reviewer and the program committee whether they would stick to these standards more broadly. My extrapolation for "fairness" from the reviewer's comments is that papers that propose new algorithms but demonstrate capabilities only on tasks that can be solved with simple look-up approaches or linear policies should be automatically down-weighted heavily. When new tasks are used, naturally they must be put to the same test to assess if simpler approaches and architectures can solve them. This is unfortunately not the case in the community and peer-review process. Does the reviewer or PC have any suggestions on how to normalize these issues?

---

> ### Author Response · Authors · 2018-01-05
> **author response**
>
> Thanks for pointing out earlier works! We should have and will definitely cite these work later in a revision.
>
> Indeed those works you pointed out evaluated their approaches on a broader set of tasks, but we would like to point out that the five tasks tested in our submission are indeed popular and widely used. It is in our plan to broaden a set of target tasks for which we evaluate these simple nearest neighbour based policies, and we will release our code to make it easier for anyone in the community to evaluate any new (or existing) task with this minimal approach to assess its difficulty more easily in the future.

---

> > ### Comment · AnonReviewer2 · 2018-01-12
> > **response**
> >
> > Thanks for the author's response. As with the other reviewers, I continue to believe this is more suited for a workshop submission.
> >
> > As I cited in my review (and hopefully this also addresses the follow-up comment), I don't believe there are recent, accepted papers which only use these simple tasks (except for some theory focused papers). The fact that many empirical results use some simple tasks is true, but they also test against a number of other more complex tasks, blunts the primary argument of this work so I will leave my rating.

---

### Official Review · AnonReviewer1 · 2017-11-29
**Well-written but nothing particularly new.**

**Rating:** 3
**Confidence:** 5

**Review:**

This paper presents a nearest-neighbor based continuous control policy.  Two algorithms are presented: NN-1 runs open-loop trajectories from the beginning state, and NN-2 runs a state-condition policy that retrieves nearest state-action tuples for each state.

The overall algorithm is very simple to implement and can do reasonably well on some simple control tasks, but quickly gets overwhelmed by higher-dimensional and stochastic environments.  It is very similar to "Learning to Steer on Winding Tracks Using Semi-Parametric Control Policies" and is effectively an indirect form of tile coding (each could be seen as a fixed voronoi cell).  I am sure this idea has been tried before in the 90s but I am not familiar enough with all the literature to find it (A quick google search brings this up: Reinforcement Learning of Active Recognition Behaviors, with a chapter on nearest-neighbor lookup for policies: https://people.eecs.berkeley.edu/~trevor/papers/1997-045/node3.html).

Although I believe there is work to be done in the current round of RL research using nearest neighbor policies, I don't believe this paper delves very far into pushing new ideas (even a simple adaptive distance metric could have provided some interesting results, nevermind doing a learned metric in a latent space to allow for rapid retrainig of a policy on new domains....), and for that reason I don't think it has a place as a conference paper at ICLR.  I would suggest its submission to a workshop where it might have more use triggering discussion of further work in this area.

---

> ### Author Response · Authors · 2018-01-05
> **author response**
>
> Thanks for pointing out some earlier works! We will cite those and probably more earlier works later in a revision.
>
> We'd like to re-emphasize that the nearest neighbour policy itself is not the point of our submission. Indeed, we did design two particular instantiations of a nearest neighbour based policy family, but the main goal was to see whether these popular existing tasks are worth benchmarks for assessing increasingly many variants of sophisticated neural net based policy algorithms, as we stated in the conclusion (though, we agree we should make it much clearer): "the perceived difficulty of these benchmark problems has largely been dominated by optimization difficulty rather than the actual, underlying difficulty of the environment and task" and "we conclude that more challenging tasks, or more diverse settings of existing benchmark tasks, are necessary to properly assess advances."

---

### Public Comment · ~Michiel_van_de_Panne1 · 2017-11-27
**Interesting work**

I enjoyed reading the paper!
One caveat with the given approach is that the distance metric becomes very important.
As the motion tasks become more complex, it may require custom distance metrics
for different motion phases or state-space regions.

I believe that the general idea has connections to "habit based learning", i.e., see
  Habits, action sequences and reinforcement learning (2012)
  https://pdfs.semanticscholar.org/ed15/6c39a0d3a5f58660b571decbf3f46da5d752.pdf

See also the 2012 paper "Optimal isn't good enough" by Loeb,
which places an emphasis on related ideas of memory-based lookup "learning"
(and an alternate philosophical point of view to optimization).

Lastly, the following papers demonstrate the efficacy of simple nearest-neighbor control policies,
using only 6-20 points to represent the entire control policy.  Caveat:  this is more of a parametric policy,
given that policy search is used to optimize these small set of "representative states and actions".

http://www.cs.ubc.ca/~van/papers/2005-icra-steering.pdf
http://www.cs.ubc.ca/~van/papers/2005-icra-walking.pdf

best wishes with this work.
Michiel

---

### Decision · Program_Chairs · 2018-01-29
**ICLR 2018 Conference Acceptance Decision**

**Decision:**

Reject

**Comment:**

Evaluating simple baselines for continuous control is important and nearest neighbor search methods are interesting. However, the reviewers think that the paper lacks citation and comparison to some prior work and evaluation on more challenging benchmarks.